# Effects of Early Nutrition and Sanitary Conditions on Oral Tolerance and Antibody Responses in Broiler Chickens

**DOI:** 10.3390/vetsci7040148

**Published:** 2020-10-01

**Authors:** Maarten S. Hollemans, Ger de Vries Reilingh, Sonja de Vries, Henk K. Parmentier, Aart Lammers

**Affiliations:** 1Coppens Diervoeding B.V., P.O. Box 79, NL-5700AB Helmond, The Netherlands; 2Adaptation Physiology Group, Wageningen University & Research, P.O. Box 338, NL-6700AH Wageningen, The Netherlands; vrie8424@planet.nl (G.d.V.R.); Henk.Parmentier@wur.nl (H.K.P.); Aart.Lammers@wur.nl (A.L.); 3Animal Nutrition Group, Wageningen University & Research, P.O. Box 338, NL-6700AH Wageningen, The Netherlands; Sonja.DeVries@wur.nl

**Keywords:** broiler chicken, early nutrition, delayed nutrition, sanitary conditions, specific antibody, natural antibody, oral tolerance

## Abstract

Greater antigenic exposure might accelerate activation and maturation of the humoral immune system. After hatch, commercial broiler chickens can have early (EN) or delayed (DN) access to nutrition, up to 72 h after hatch. The immune system of EN versus DN broilers is likely more exposed to antigens after hatch. This might contribute to activation and maturation of the immune system, but might also influence the development of oral tolerance, thereby altering later life antibody responses. We studied antibody (IgM, IgY, IgA) responses between 21 and 42 d of age in fast-growing EN and DN broilers, kept under low (LSC) or high sanitary conditions (HSC). In a first experiment (*n* = 51 broilers), we tested whether early oral exposure to bovine serum albumin (BSA) affected later life antibody responses towards BSA and a novel antigen—rabbit γ-globulin (RGG), under HSC. In a second experiment, a total of 480 EN and DN broilers were housed under either LSC or HSC, and we studied antibody responses against both BSA and RGG (*n* = 48 broilers per treatment) and growth performance. Broilers kept under LSC versus HSC, had higher antibody levels and their growth performance was severely depressed. Interactions between feeding strategy (EN versus DN) and sanitary conditions, or main effects of feeding strategy, on natural and specific antibody levels, and growth performance were not observed. Levels of IgA were elevated in EN versus DN broilers, in experiment I and in batch 2 of experiment II, but not in the other batches of experiment II. We concluded that EN versus DN contributes minimally to the regulation of antibody responses, irrespective of antigenic pressure in the rearing environment.

## 1. Introduction

Just-hatched broiler chickens can experience a prolonged delay in access to nutrition for up to 72 h (delayed nutrition; DN), especially in the case of long post-hatch transport (for review, see [1]). Broilers that received early nutrition (EN) and thus immediate provision of nutrition, post hatch onwards, showed enhanced immune activation and maturation compared to DN broilers; reviewed by [2]. This is likely caused by early exposure of the immune system to antigens derived from commensal microbiota or ingested feed [3,4]. Thus, broilers receiving EN compared to DN, are expected to be exposed to a higher and more diverse load of antigens, derived from ingested feed and subsequent microbial colonization [5,6,7]. In chicken, antibody responses towards antigens were reduced when these antigens were fed within the first 72 h after hatch (“window of opportunity”), which is known as oral tolerance [8]. Interestingly, this window of opportunity parallels with the application of different feeding strategies (EN versus DN), during the first 72 h after hatch. Hence, it could be postulated that increased antigenic exposure as a result of EN, compared to DN, might affect the development of oral tolerance, resulting in altered later life antibody responses.

Indeed, EN versus DN broilers that were housed under relatively high antigenic pressure (floor housing) had lower IgY in blood plasma and growth depression, after immunization with LPS and HuSA, as compared to broilers housed under relatively low antigenic pressure (cage housing) [9]. These authors suggested that effects of EN compared with DN on antibody responses might be affected by antigenic pressure. Studies comparing the later life immune system between EN and DN broilers under lower antigenic pressure, observed no differences among treatments [9,10]. This suggests that later life immune responses might be regulated by EN, but only when kept in a high antigenic pressure environment. Hence, we hypothesized that better regulation of antibody responses in EN compared to DN broilers, would be beneficial when the broilers are housed under high antigenic pressure. To test this hypothesis, we executed two consecutive experiments to understand the effects of feeding strategy (EN versus DN) and its interaction with antigenic pressure on later life antibody responses. The aim of experiment I was to test, under low antigenic pressure, whether EN might modulate later life antibody responses. In experiment II, we modeled contrasts in antigenic pressure by creating low (LSC) and high sanitary conditions (HSC). This allowed us to test whether the effects of feeding strategy on regulation of antibody responses and growth performance, depend on environmental antigenic pressure.

## 2. Materials and Methods

### 2.1. Experimental Designs

The experiments and respective procedures were ethically approved, according to the Dutch law under application number AVD104002016441, and were performed in climate respiration chambers (CRC) at the experimental facility. In both experiments, Ross 308 males were collected from a commercial hatchery within 1 h after hatch, to minimize age differences, and were immediately transported to the experimental facility, within 20 min, in cardboard chick boxes kept at 20 °C. All ages are expressed as biological age [11]. After arrival, the broilers received an ID neck tag, and were distributed within 2 h over the floor pens (1.1 × 1.8 m) that contained SoftCell (Agromed GmbH, Kremsmünster, Austria) as bedding material, covered with chicken paper during the first 3 d of age, to prevent litter uptake. In both experiments, light and climate settings were identical. From placement onwards, a 16-h light:8-h dark schedule was applied. Ambient temperature was set at 36 °C (55% relative humidity) at 0 d of age, and gradually reduced to 29 °C until 7 d, and then further gradually reduced to 18 °C (75% relative humidity) at 35 d. Relative humidity was set at 55% at the start of the experiment and gradually increased to 75% at 42 d. Levels of CO2 were maintained ≤2500 ppm and that of NH3 ≤ 20 ppm. In both experiments, broilers had ad libitum access to water and a commercial broiler diet, except for the DN chickens, which had no access to feed and water, during the first 72 h from placement onwards.

#### 2.1.1. Experiment I

The experiment was conducted in a single CRC, and designed as a 2 × 2 factorial approach, with antigen feeding (BSA feeding, PBS feeding) and early life feeding strategy (EN, DN) as factors. The 4 treatments were distributed over 12 floor pens, each containing 4 broilers. Three surplus broilers were distributed over the treatments, resulting in a total of 13 broilers for all treatments, except for the EN–BSA-fed broilers (*n* = 12). Body weight (BW) of the individual broilers was measured weekly. Broilers were not vaccinated in the hatchery or during the study. Commercial pelletized broiler starter (0–14 d; digestible energy (DE): 2950 kcal/kg; total lysine: 12.2 g/kg) and finisher diets (14–42 d; DE: 3000 kcal/kg; total lysine: 11.3 g/kg) were fed.

#### 2.1.2. Experiment II

The experiment was designed as a 2 × 2 factorial approach with early life feeding strategy (EN, DN) and sanitary conditions (LSC, HSC) as factors. Low sanitary conditions were induced by the introduction of used litter from commercial broiler farms (see Section 2.2). The experiment was executed in 3 consecutive batches, to account for variations in antigenic pressure due to differences in health status of litter-donating farms, as previously demonstrated in a pig model (Van der Meer et al., 2016). Broilers (parent stock age: batch 1: 31 w, batch 2: 33 w, batch 3: 48 w) were housed in either an LSC or an HSC CRC, each containing 8 floor pens. Both CRC were completely identical in their set-up and were controlled for identical climate conditions (temperature, humidity, CO2, NH3) (Van der Meer, 2017). Body weight and feed intake were measured weekly until 33 d of age to calculate the average daily gain (ADG), feed intake (ADFI), and feed conversion ratio (FCR). Broilers were vaccinated against Newcastle disease at 3 d of age, but this was accidentally omitted in batch 1. Broilers received no other vaccinations in the hatchery or during the study. Commercial pelletized broiler starter (0–7 d; DE: 2850 kcal/kg; total lysine: 11.8 g/kg), grower (7–28 d; DE: 2900 kcal/kg; total lysine: 11.2 g/kg), and finisher diets (28–35 d; DE: 2950 kcal/kg; total lysine: 10.7 g/kg) were fed. The grower diet contained decoquinate (0.05 g/kg; Deccox 6%, Zoetis, Capelle aan den IJssel, The Netherlands).

### 2.2. Sanitary Conditions

In experiment I, the broilers were kept under HSC from placement onwards, as follows. The HSC chamber was cleaned with water and disinfected (Halamid, Veip Disinfectants, Wijk bij Duurstede, The Netherlands), following the manufacturer’s instructions, and over-pressurized (100 ± 5 Pa) to prevent the influx of external pathogens. A strict hygiene protocol (consisting of showering, cleaning and disinfection boots, wearing gloves and hairnet, and minimal pen entrance) was maintained. All procedures, except for dissection, were performed inside the chamber, to prevent introduction of novel antigens. In experiment II, all broilers were kept under HSC (similar to experiment I) until 3 d of age, after which the contrasts in sanitary conditions were made as follows. The HSC broilers were kept under HSC until the end of the experiment. Low sanitary conditions were induced as follows—the LSC chamber was under-pressurized (−65 ± 5 Pa) and no hygiene protocol was maintained. Coveralls and boots were not cleaned and disinfected. Seven days before onset of each respective batch, litter was obtained from 3 commercial broiler farms with flock age of at least 35 d. The litter was collected all at once, during the cleaning of the broiler houses, and after arrival at the experimental facility, the litter was pooled by weight and stored in 8–10 kg portions at 4 °C. From 3 d of age onwards, one portion (8–10 kg) of homogenized used litter was distributed in each pen, every 4 d.

### 2.3. Induction of Oral Tolerance and Immunizations

From placement until 3 d of age, broilers were orally fed with either BSA to induce tolerance (V = 0.25 mL; 100 mg/mL; Sigma Aldrich CO, St. Louis, MO, USA) or phosphate buffered saline (PBS) as a control, every 12 h. In experiment I, BSA or PBS was administered by pipetting the volume in the beak of each broiler. In experiment II, BSA was administered by oral gavage in the esophagus via a blunted needle. An intratracheal dose (V = 0.5 mL) containing 0.5 mg BSA and 0.5 mg RGG (rabbit γ-globulin, Sigma) was given at 21 and 22 d of age (experiment I; all broilers) or 24 d of age (experiment II; 2 broilers per pen), via a blunted needle, as a secondary immunization.

### 2.4. Sample Collection

In experiment I, blood was collected at 21, 24, 28, 35, and 42 d (0, 3, 7, 14, and 14 d post immunization (p.i.)) from the wing vein of all broilers in heparinized tubes, and subsequently centrifuged (12,000× *g*, 5 min) to obtain plasma. In experiment II, blood was collected at 14, 24, 29, and 33 d (−10, 0, 5, and 9 d p.i.) from 1 broiler per pen in tubes, incubated for 2 h at 4 °C, and subsequently centrifuged (12,000× *g*, 5 min) to obtain serum. Bile was collected from all broilers either at 42 d of age (experiment I) or 33 and 34 d of age (experiment II), as follows. After euthanizing the broiler by a percussive blow on the head, the bile bladder was located, and the contents were collected with a syringe and needle, and placed in tubes. The spleens were located, collected, and weighed. Plasma, sera, and bile were stored at −20 °C for further analyses. In experiment II, after euthanizing the broiler by decapitation, the whole right cecum with contents was collected within 6 h after hatch and at 1, 2, 3, and 7 d of age, from one broiler per pen. The cecum was weighed, placed in sterile 0.9% NaCl, and homogenized with a tissue homogenizer (IKA T50, IKA-Werke, Staufen, Germany). Afterwards, the homogenate was diluted 1:1 in sterile 15% glycerol, and stored at −80 °C, pending further analyses.

### 2.5. Bacterial Load in Cecum

All procedures were performed under aseptic conditions. The specific culture media MacConkey agar (MAC; CM0007), De Man, Rogosa, and Sharpe agar (MRS; CM0359), and blood agar (BA; OXOIPB5039A) were prepared according to the instructions of the supplier (Oxoid Microbiology Products, Hampshire, United Kingdom). After pouring the specific culture media, the homogenates were thawed at 20 °C and singular, 10-fold serial dilutions in sterile PBS were spread on the agar plates (100 µL/plate). Plates were incubated upside down at 37 °C in an incubator, for 36 h (BA, MAC) or 48 h (MRS). After incubation, the number of colony forming units (CFU) were counted and expressed per gram of collected cecal tissue.

### 2.6. Antibody Levels

Two-step, indirect, enzyme-linked immunosorbent assays (ELISA) were performed to measure IgM and IgY levels in plasma or serum, and IgA levels in bile, binding either BSA or rabbit γ-globulin (RGG). Flat-bottomed, 96-well medium binding plates (Greiner Bio-One) were coated with either 2.5 µg/mL BSA or 2.5 µg/mL RGG in 100-µl coating buffer (5.3 g/L Na_2_CO_3_, 4.2 g/L NaHCO_3_; pH 9.6), and incubated overnight at 4 °C. After washing with tap water containing 0.05% Tween^®^ 20, plates were filled with 100 µl dilution buffer (PBS (10.26 g/L Na_2_HPO_4_·2H_2_O, 2.36 g/L KH_2_PO_4_, 4.5 g/L NaCl; pH 7.2) containing 0.5% normal horse serum and 0.05% Tween^®^ 20). Prediluted samples (1:10 in dilution buffer) were further diluted to 1:40, 1:160, 1:640, and 1:2560, and a standard positive control was included, in duplicates, from the pooled plasma, serum, or bile, for each day. Plates were incubated for 1.5 h at 20 °C. After washing, the plates were incubated with 1:20.000-diluted goat-anti-chicken IgA, goat-anti-chicken IgM, or goat-anti-chicken IgY, all labelled with horse radish peroxidase (Bethyl Laboratories Inc, Montgomery, AL, USA), for 1.5 h at 20 °C. After washing, 100 µL substrate buffer (containing reverse osmosis purified water, 10% tetramethylbenzidine buffer (15.0 g/L sodium acetate, 1.43 g/L urea hydrogen peroxide; pH 5.5) and 1% tetramethylbenzidine (8 g/L DMSO) were added. After 15 min of incubation at 20 °C, the reaction was stopped with 50 µL of 1.25 M H_2_SO_4_ solution. The extinctions were measured with a Multiskan GO (Thermo scientific, Breda, The Netherlands) at 450 nm. Titers were expressed as log2 values of the dilutions that gave an extinction closest to 50% of E_max_, where E_max_ represents the highest mean extinction of the standard positive.

### 2.7. Statistical Analyses

Data were processed, analyzed, and presented using R version 3.6.1 [12]. Linear mixed models were established with the nlme package version 3.1–140 [13]. Model residuals of the linear models were tested to verify the assumptions of normality and homogeneity by QQ-plots and residual plots. Differences among means with *p* ≤ 0.05 were considered to be statistically significant, and *p* ≤ 0.10 were considered to be tendencies. All data are presented as (back-transformed) estimated marginal means with standard errors, unless specified otherwise.

#### 2.7.1. Experiment I

General linear models were established to estimate the effects of BSA feeding (PBS fed, BSA fed), feeding (DN, EN), and their interaction on antibody levels, binding either BSA or RGG in plasma (IgM, IgY) on 0, 14, 21 d p.i., fold change (ratio between IgM or IgY levels at 7 versus 0 d p.i.), IgA levels in bile (21 d p.i.), BW (0, 21, 24, 42 d), and ADG (0–24, 21–24, 24–42, 0–42) with individual broiler within the pen as the experimental unit, and pen as the random effect. Logarithmic transformation was applied to normalize the residuals on all titer and performance data, except for the IgA binding BSA and RGG.

#### 2.7.2. Experiment II

General linear models were established to estimate the effects of sanitary condition (LSC, HSC), feeding (DN, EN), and their interaction on antibody levels, binding either BSA or RGG in serum (IgM, IgY) on −10, 0, 7 d p.i., and fold change (ratio 5:0 levels d p.i.; IgM, IgY), with pen as the experimental unit. Batch (1, 2, 3) and its interaction with treatment effects was used as the blocking factor. Comparable models were used to analyze the levels of bile and spleen weights among treatments. As bile and spleens were collected on 2 consecutive days (10, 11 d p.i.), sampling day was added as a blocking factor. Day was found to not be significant and therefore was eliminated from the models. Effects of treatments on BW (3, 33 d), ADG (3–14; 14–28; 28–33 d), and FCR (3–14; 14–28; 28–33 d) were estimated with general linear models, using sanitary condition, feeding, and their respective interaction as fixed effects, with pen as the experimental unit. Fixed effects of treatments (EN, DN) on bacterial colonization in ceca were analyzed using the non-parametric Kruskal-Wallis test, separately for each day (0, 1, 2, 3, 7). For day 7, effects of sanitary condition (HSC, LSC) were also analyzed. Logarithmic transformation was applied to normalize the residuals of all titer and performance data. Interactions between batch and other main effects (feeding, sanitary condition, feeding * sanitary condition) were tested but excluded from the model if *p* > 0.10.

## 3. Results

### 3.1. Experiment I: Effects of Early Antigenic Exposure on Later Life Antibody Responses

In experiment I, we studied the effects of BSA feeding and feeding strategy on antibody levels in blood plasma, prior to immunization (0 d p.i.; natural antibody (NAb) levels [14,15]), and specific antibodies (SpAb), after immunization with BSA (7, 14, 21 d p.i.). Both NAb and SpAb levels binding BSA were not affected by the interaction between BSA-feeding and EN, or the main effects of EN (Figure 1A). At 0 d p.i., a tendency (*p* = 0.10) for the IgY binding BSA was found, indicating lower levels of NAb binding BSA, in BSA-fed broilers (1.3 ± 0.3), compared to PBS-fed broilers (2.1 ± 0.5). At 14 and 21 d p.i., levels of both IgM and IgY SpAb binding BSA were reduced (*p* ≤ 0.05) in the BSA-fed groups, compared to the PBS-fed group.

Specific antibody responses towards RGG were measured to test whether BSA feeding at 0 to 3 d of age induced specific tolerance towards BSA or not. For this purpose, the broilers were fed with BSA, but not with RGG, and were subsequently immunized with both BSA and RGG at 21 d of age. This approach enabled us to also investigate whether EN versus DN affected the regular antibody responses. No main effects (*p* > 0.10) of BSA feeding on the antibodies binding RGG (Figure 1B) were present, indicating that indeed antigen feeding in early life is required to modulate antibody responses. No interaction effects or main effects of feeding strategy were present on the antibodies binding RGG. However, a tendency (*p* = 0.10) at 21 d p.i. indicated higher levels of IgY SpAb binding RGG in the BSA-fed broilers (9.1 ± 0.4), compared to the PBS-fed broilers (9.9 ± 0.5). The fold-change at 7 d p.i. was lower (*p* = 0.05) for the IgY binding RGG, in broilers receiving EN (11.8 ± 1.9), as compared to DN (16.7 ± 2.6).

Levels of IgA binding BSA were compared between the BSA and PBS-fed broilers in bile collected at 21 d p.i. Although no interaction between BSA feeding and EN was found (*p* = 0.47), both BSA-feeding (*p* = 0.05) and EN (*p* = 0.05) resulted in higher IgA levels binding BSA (Table 1). We confirmed that higher levels of IgA were BSA-specific, as IgA binding RGG was not affected by any treatments.

No effects of BSA-feeding on BW and ADG were found (*p* > 0.05), and therefore, the BSA and PBS-fed groups were combined in Table 2. Body weight was greater (*p* ≤ 0.05) upon 24 d of age, and slaughter weight (d 42) tended to be greater (Δ = 216 g; *p* = 0.09) in EN, compared to the DN group. Average daily gain was greater in EN, compared to the DN groups, between 0–21 d (*p* = 0.001) and 21–24 d (*p* = 0.07).

### 3.2. Experiment II: Effects of Early Life Feeding Strategy on Antibody Responses under Different Sanitary Conditions

#### 3.2.1. Bacterial Colonization

Bacterial load (CFU/g cecal tissue) for cultivable aerobic bacteria on BA, MAC, and MRS agar is presented in Figure 2. We observed relatively great within-group variation in colonization, irrespective of feeding strategy, on all agars directly after hatch, and at 1 d of age. At 1 d of age, some individual broilers appeared to be already colonized, while others were not. Within-group variation declined and the aerobic counts stabilized between 10^7.5^ and 10^10^ CFU, from 2 d of age onwards for BA and MAC, and from 3 d of age for MRS. At 3 d of age, tendencies (all *p* ≤ 0.10) were present, suggesting more CFU on the BA and MAC agar in the DN broilers, compared to the EN broilers. At 7 d of age, no differences between sanitary conditions were present (data not presented).

#### 3.2.2. Antibody Levels and Responses

Similar to experiment I, we measured the levels of Nab- and SpAb-binding BSA or RGG in BSA-fed (0–3 d of age) broilers, after BSA and RGG immunization. We observed no interaction between the feeding strategy and sanitary conditions on the NAb and SpAb levels (IgM and IgY) binding BSA (Figure 3A), or RGG (Figure 3B). IgY-binding RGG tended to be lower (Δ = 0.7; *p* = 0.09) at 14 d/−10 d p.i. in EN, compared to the DN groups. Broilers housed under LSC, compared to HSC, showed increased levels of both NAb and SpAb. We observed increased (*p* ≤ 0.05) NAb levels binding BSA or RGG under LSC, at 14 d/−10 d p.i. (IgM) and 24 d/0 d p.i. (IgM, IgY). Specific antibody levels binding RGG (IgM) and BSA (IgM, IgY) were increased (*p* < 0.05) in the LSC groups. In LSC, compared to the HSC groups, we observed a reduced fold-change (within 7 d p.i.) of BSA-binding IgM (*p* = 0.01), but not IgY (Table 3). With regard to the antibodies binding RGG, fold-change was higher in HSC, compared to LSC for IgM (*p* < 0.01), and a tendency was found for IgY (*p* = 0.07).

We measured the biliary levels of IgA binding BSA or RGG to study the effects of feeding strategy (EN versus DN) and sanitary conditions (LSC versus HSC), which after release in the intestinal tract likely played a role in maintenance of mucosal homeostasis [5,6,7]. Biliary IgA binding BSA at 11 d p.i. was not affected by the interaction between feeding strategy and sanitary conditions. However, in batch 2, a tendency for an interaction (*p* = 0.06) between batch and feeding strategy indicated higher (Δ = 0.9) levels of IgA binding BSA in EN, compared to the DN groups. Immunoglobulin A levels binding RGG were consistently higher (Δ = 0.3; *p* = 0.03) over all batches in EN compared to the DN groups. With regard to sanitary conditions, the LSC groups had higher (*p* < 0.001) IgA levels binding BSA (Δ = 1.7) and RGG (Δ = 1.3; Table 4).

#### 3.2.3. Growth Performance

As we conducted experiment II in 3 consecutive batches, in which the litter of different broiler farms was introduced to obtain LSC, we tested the interactions between batch and sanitary conditions, which might indicate differences in antigenic pressure among batches in the LSC groups. Interactions between batch and sanitary conditions were present (*p* ≤ 0.05) for BW, ADG, and FCR. In general, the greatest differences in BW between LSC and HSC were observed at 33 d of age, with the greatest effect size in batch 1 (Δ = 623 ± 62.7 g), followed by batch 3 (Δ = 419 ± 62.7 g), and finally batch 2 (Δ = 301 ± 62.7 g).

Broiler growth performance was measured in this experiment to study whether EN compared with DN broilers, were better able to cope with high antigenic pressure or not (LSC versus HSC). Average daily feed intake between 3–14 d of age was greatest in the EN broilers housed under HSC (37 ± 0.7 g) and lowest in the DN broilers housed under LSC (21 ± 0.7 g) (feeding strategy × sanitary conditions, *p* = 0.03). No interaction between the feeding strategy and sanitary conditions was found for BW, ADG, and FCR (Table 5).

At 3 d of age (after 72 h delay in nutrition), we observed greater (Δ = 30.2 g; *p* < 0.001) BW in EN compared with DN groups. Differences in BW between the EN and DN groups persisted until 33 d of age (Δ = 210 g). Throughout the complete experiment, ADFI was affected by feeding strategy (*p* ≤ 0.001), resulting in greater (Δ = 14 g) ADFI between 3 and 33 d of age. We observed reduced (Δ = 0.06; *p* = 0.01) feed conversion ratio in the EN group compared to the DN groups, between 14 and 28 d. Between 3 and 33 d of age, FCR was higher (Δ = 0.04; *p* ≤ 0.001) in the EN group, compared to the DN groups. As large differences in final BW were present between the EN and DN, we corrected FCR to a standardized BW of 2000 g at 33 d of age [5,6,7]. Standardized FCR was, however, unaffected by the feeding strategy.

The introduction of LSC from 3 d of age onwards resulted in a reduction (Δ = 448 g; *p* < 0.001) of BW at 33 d of age in LSC, compared to the HSC groups. Between 3–33 d of age, ADG was greater (Δ = 9 g; *p* < 0.001) in the EN group, as compared to the DN group, and it was lower (Δ = −17 g; *p* < 0.001) in LSC, compared to the HSC group. Feed conversion ratio was unaffected by sanitary conditions for up to 14 d, but FCR was increased in LSC, compared to the HSC groups, at 14–28 d (Δ = 0.06; *p* < 0.01) and 28–33 d (Δ = 0.06; *p* = 0.03). Between 3 and 33 d of age, FCR was higher (Δ = 0.07; *p* < 0.001) in LSC, compared to HSC. Feed conversion ratio standardized to 2000 g of BW at 33 d was greater (Δ = 0.11; *p* < 0.001) in LSC, compared to the HSC groups.

## 4. Discussion

We studied whether antibody responses of the broilers are influenced (either increased or decreased) by early life feeding strategy (EN versus DN) and sanitary conditions (LSC versus HSC), in two consecutive experiments. First, we studied whether feeding BSA during the first 3 d of life (“window of opportunity”) affects later life antibody responses and its interaction with feeding strategy, through the development of oral tolerance. Second, we tested the effects of EN on (regulation of) antibody immune responses and growth performance, under either HSC or LSC. We observed no effects of the feeding strategy on antibody responses, irrespective of sanitary conditions. Broilers kept under LSC, as compared to HSC, showed higher levels of natural antibodies (NAb) and a smaller fold-change of specific antibodies (SpAb) after BSA and RGG immunization, and a reduced growth performance.

### 4.1. Aerobic Bacterial Colonization after Early Nutrition

Accelerated maturation of the immune system after EN as compared to DN, was suggested to be caused by enhanced bacterial colonization of the intestinal tract [5,6,7]. Short-term effects of EN compared to DN, were found on the intestinal bacterial load [16]. Data on the effects of EN compared to DN, on intestinal bacterial composition, suggest temporal differences in the ileal bacterial composition, up to 9 d of age [16]. To relate these effects to antibody responses, we analyzed the colonization dynamics of culturable aerobic bacteria in ceca, between EN and DN broilers, in the first week post-hatch (experiment II).

The intestinal tract of broilers was rapidly colonized by bacteria, within the first days after hatch and the total number of colonizing bacteria stabilized around 3 d of age, which was in accordance with previous research reviewed by [17]. Relatively high within-treatment group variation of up to 2 (BA, MAC) or 3 (MRS) d of age, might suggest divergent colonization patterns among individuals. We observed no significant effects of feeding strategy (EN versus DN) up to 3 d of age on bacterial colonization, suggesting that EN, compared to DN, minimally affected aerobic bacterial load. In EN, compared to DN broilers, the aerobic bacterial load tended to be lower at 3 d, which was also previously observed by others [5]. Speculatively, this might reflect earlier replacement of aerobes (*E. coli*, *Enterococcus* spp.) by obligate anaerobic species, after EN, as suggested before [5].

### 4.2. Early-Life BSA Feeding Lowers Later-Life Antibody Responses Towards BSA under High Sanitary Conditions

In experiment I, we first tested whether early-life oral exposure towards antigens affected later-life antibody responses. Therefore, BSA or PBS (negative control) was orally administered during the first 3 d of age. At 21 d of age, broilers received an i.t. BSA immunization, following procedures derived from the oral tolerance model in laying hens [8]. Although broilers are known to be immunologically different from laying hens [18,19,20], our study demonstrated that the mechanism of oral tolerance also exist in broilers. Thus, our findings confirmed that after immunization at 21 d in broilers, antibody responses are lowered towards antigens that were fed in the first 3 d after hatch [8,21,22]. We also investigated whether BSA feeding in early-life (0–3 d of age) affect SpAb responses to a non-BSA related antigen. Therefore, broilers were immunized i.t. with RGG at 21 d of age, to test whether antibody responses differed between the BSA and PBS-fed groups. We observed that systemic IgM and IgY (Figure 1B) and biliary IgA (Table 1) antibodies binding RGG, were unaffected (*p* ≥ 0.05) by BSA feeding. Thus, early-life oral antigen exposure affected antibody responses in a specific fashion towards the specifically fed antigen, and not to novel, unrelated antigens. Effects of BSA feeding on IgA responses in bile were studied at 21 d p.i., as an indicator for immune regulation—IgA is present at the mucosal surfaces and contributes to immune homeostasis by binding antigens in the intestinal lumen and coating intestinal bacteria [23,24,25]. Our finding that BSA feeding results in higher levels of IgA binding BSA, suggests that early-life antigen exposure also affects IgA responses. In summary, we demonstrated that the specific IgA response is increased, and specific IgM and IgY responses are lowered, by oral exposure to antigen (BSA), during the first 3 d of age.

The main objective of experiment I was to test whether the feeding strategy (EN versus DN) affected later-life antibody responses towards BSA and RGG in BSA-fed broilers. We observed no effects of feeding strategy on systemic antibody responses towards BSA or RGG, indicating that immediate provision of nutrition after hatch (EN) did not interfere with antibody responses at later-life. Irrespective of BSA feeding, we observed higher levels of biliary IgA-binding BSA in EN broilers, suggesting that EN contributes to a more anti-inflammatory immune status in EN broilers. We would like to emphasize that experiment I was conducted under HSC. Limited antigenic pressure in experiment I might explain the lack of differences in IgM and IgY responses between EN and DN broilers, as less regulation of immune responses was required. Therefore, we designed a second experiment where the broilers were kept under LSC or HSC.

### 4.3. Introduction of Low Sanitary Conditions

To model contrasts in antigenic pressure in experiment II, we introduced either LSC and HSC from 3 d of age onwards, according to a pig model [26,27]. As litter from 3 different broiler farms per batch was used in LSC, we considered the obtained antigenic pressure to be illustrative for the range of antigens found in typical commercial broiler farms. Throughout the experiment, growth performance of broilers was depressed in LSC compared to the HSC groups. Relative spleen weights tended to be greater (*p* = 0.09) at 33 d of age in LSC (0.26 ± 0.09), compared to the HSC broilers (0.11 ± 0.09), indicating greater activity of the immune system [28]. From these data, we conclude that LSC was successfully induced. Batch and batch × treatment effects were present in this broiler model, which was in line with the pig model [29]. We observed interactions (*p* ≤ 0.05) between batch and sanitary conditions for IgM binding BSA or RGG at 14 d/−10 d p.i., as well for BW (33 d of age), ADG (all phases), FCR (3–14 d), and standardized FCR. The observed batch effects did not change the direction, but only changed the magnitude of the observed effects. Hence, the batches were not analyzed separately.

### 4.4. Antibody Responses and Performance after Early Nutrition under Different Sanitary Conditions

In experiment II, we aimed to compare antibody responses and growth performance of the EN and DN broilers kept under either LSC or HSC. We could not confirm our hypothesis that EN versus DN broilers housed under LSC, differed in growth performance or antibody responses after BSA and RGG immunization. This was unexpected, as lower IgY responses and enhanced performance in EN compared to the DN broilers were reported in a previous study [30]. In that study, however, broilers were immunized at one time-point, with a combination of the model antigens HuSA and LPS, at 28 d of age, while in our study, LSC broilers were continuously subjected to high antigenic pressure from 3 d of age onwards. In experiment II, we observed higher BSA-specific IgA levels in EN compared to DN broilers in batch 2, but not in other batches (batch × feeding strategy interaction; *p* = 0.06). Levels of IgA-binding RGG were found to be higher in EN compared to the DN broilers, which was in contrast with experiment I, where we observed no effects of feeding strategy. As the titer differences (Δ = 0.3) between the EN and DN groups were relatively small in experiment I, it was unclear if the observed effects were biologically relevant. In summary, we concluded that EN compared to DN, minimally affected antibody responses, regardless of sanitary conditions, although the effects of feeding strategy on biliary IgA (both BSA and RGG) remained debatable.

### 4.5. Low Sanitary Conditions Increased Antibody Levels

Broilers housed under LSC from 3 d onwards, appeared to adapt towards the higher antigenic pressure from 14 d of age onwards. Levels of Nab-binding BSA or RGG before immunization with BSA and RGG, were higher in LSC, compared to the HSC broilers, at 14 d (−10 d p.i.; IgM) and 24 d (0 d p.i.; IgM, IgY) of age. Higher levels of natural IgM and the interaction between batch and sanitary condition on IgM levels, suggest that IgM NAb levels were affected by antigenic pressure from, at least, 14 d of age. Broilers kept under HSC, compared to LSC, showed a higher fold-change of IgM-binding BSA or RGG (*p* ≤ 0.01), and IgY-binding RGG (*p* = 0.07), after immunization. It is tempting to speculate that the higher levels of NAb and lower fold-change of SpAb after immunization in LSC broilers might indicate different defense strategies of the immune system. Whereas LSC broilers might depend on a less specific antibody (NAb) repertoire as a first line of defense, broilers under HSC need to generate higher specific immune responses (SpAb) after antigenic stimulation. Higher NAb levels, and lower fold-change of specific antibodies, were reported in layer hens that were continuously immunized with other antigens, compared to the non-immunized hens [30]. The observed higher levels of biliary IgA-binding BSA or RGG (both *p* < 0.001) in LSC, compared to HSC broilers, indicate increased protection of mucosal surfaces in LSC broilers [23,24]. This suggests that the immune system prevents innocuous responses to antigens during higher antigenic pressure through increasing IgA levels.

## 5. Conclusions

We demonstrated that EN, compared to DN, marginally affected the numbers of culturable aerobic bacteria in the cecum. Early nutrition resulted in higher biliary levels of IgA, but systemic IgM and IgY responses were not affected. Irrespective of feeding strategy, LSC compared to the HSC broilers were found to show depressed growth performance, and the IgM and IgY NAb levels were increased. As there was no interaction between the feeding strategy and sanitary conditions, we concluded that EN versus DN minimally contributed to the regulation of antibody responses, irrespective of antigenic pressure in the rearing environment.

## Figures and Tables

**Figure 1 vetsci-07-00148-f001:**
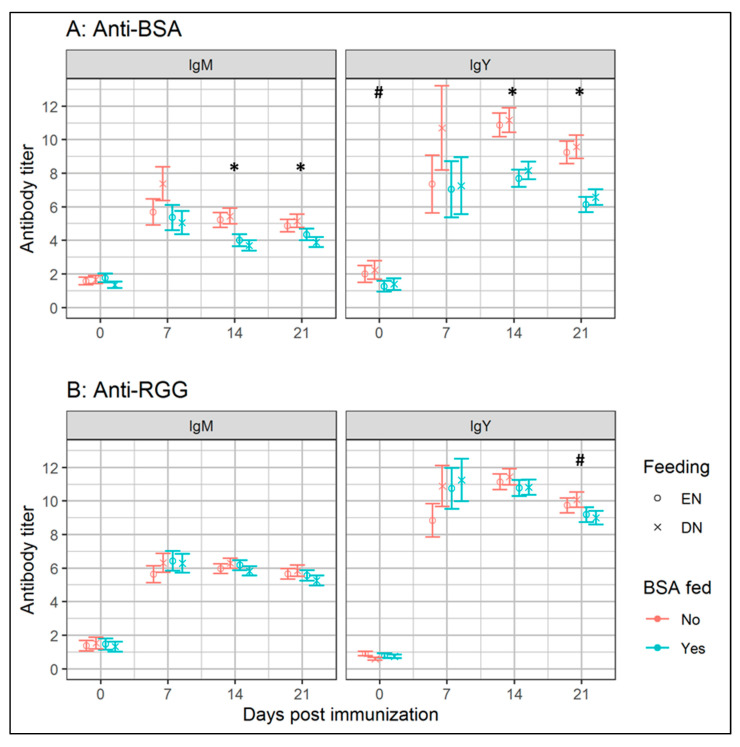
IgM and IgY binding bovine serum albumin (BSA; panel **A**) or rabbit γ-globulin (RGG; panel **B**) in the blood plasma of broiler chickens fed with BSA during the first 3 d of age, or not, and receiving either delayed (DN) or early nutrition (EN) in experiment I. Broilers were intratracheally immunized with BSA and RGG at 21 d of age (0 d p.i.). Data are presented as estimated marginal means, with the error bars representing the standard error. Within different ages, differences between means of BSA and PBS-fed groups are indicated by * (*p* ≤ 0.05) or # (*p* ≤ 0.10). *n* = 13 replicate broilers, housed in groups of 4–5 broilers per pen for all groups, except for the EN-BSA-fed, where *n* = 12 replicate broilers.

**Figure 2 vetsci-07-00148-f002:**
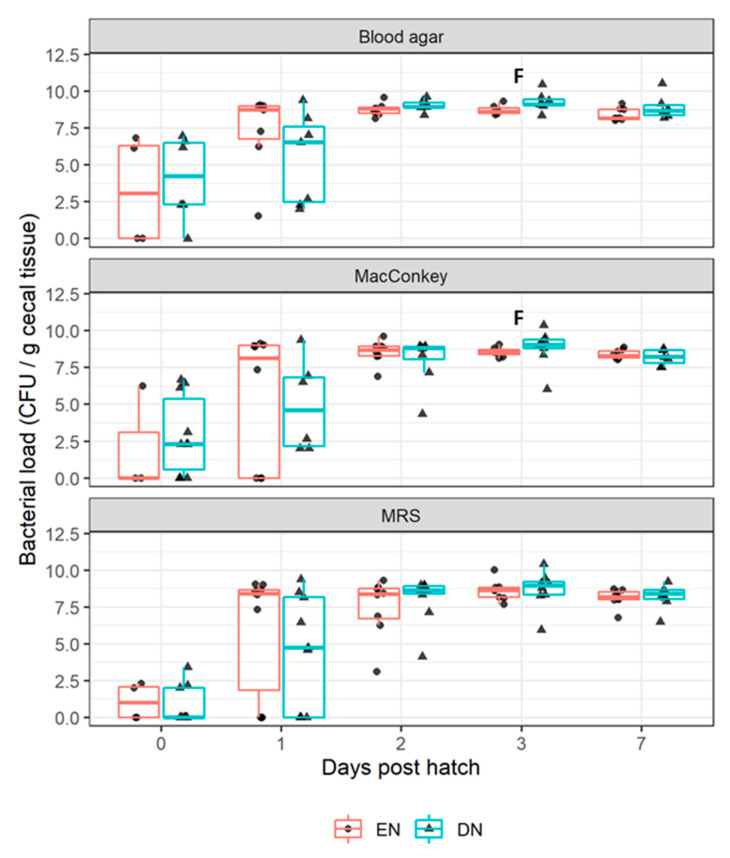
Number of aerobic cultured bacteria on 3 different growth media (blood, MacConkey, and De Man, Rogosa, and Sharpe (MRS) agar) expressed in log 10 CFU (colony forming units) per gram of cecal tissue, during the first week post-hatch, in the cecum of broiler chickens receiving delayed (DN) or early nutrition (EN) in experiment II. Colony forming units from individual broiler (dots and triangles) are summarized by boxplots. Tendencies indicating age differences between feeding groups within age are indicated by F (*p* ≤ 0.10). The number of replicate broilers varied between 3–10 per age and feeding group. The horizontal line within each boxplot represents the median and the whiskers, extended to the 1.5 × interquartile range.

**Figure 3 vetsci-07-00148-f003:**
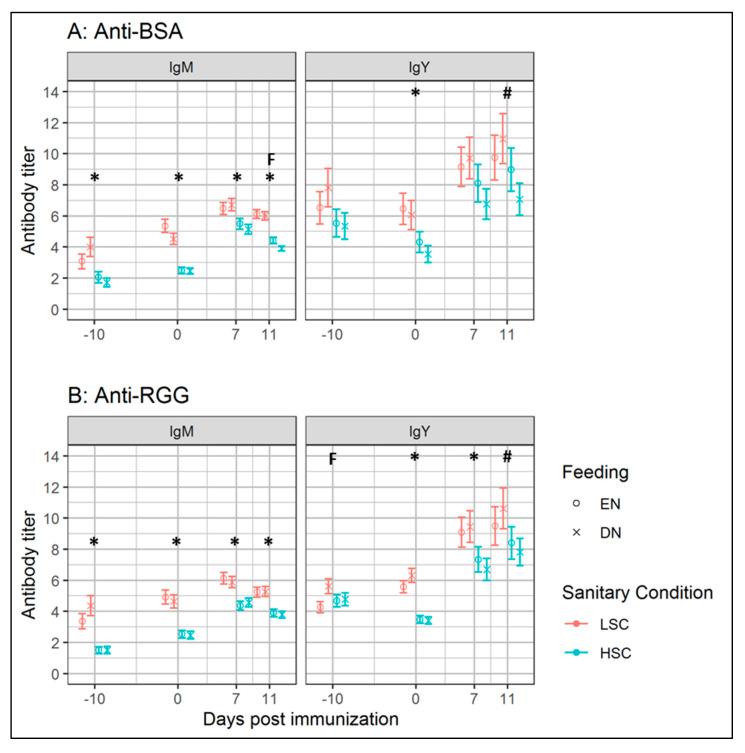
IgM and IgY-binding bovine serum albumin (BSA; panel **A**) or rabbit γ-globulin (RGG; panel **B**) in blood serum of broiler chickens fed with BSA, during the first 3 d of age, receiving delayed (DN) or early nutrition (EN), and kept under either high sanitary conditions (HSC) or under low sanitary conditions (LSC) in experiment II. Broilers were intratracheally immunized with BSA and RGG, at 24 d of age (0 d p.i.). Data are presented as estimated marginal means, with error bars. Within different ages, differences between sanitary condition groups are indicated by * (*p* ≤ 0.05) or # (*p* ≤ 0.10), and tendencies between feeding groups are indicated by F (*p* ≤ 0.10). *n* = 12 replicate broiler housed in groups of 5 broilers per pen for all groups, except for 7 and 11 d p.i., where *n* = 10 replicate broilers for all groups, with the exception of DN–HSC (*n* = 12 broilers).

**Table 1 vetsci-07-00148-t001:** IgA binding bovine serum albumin (BSA) or rabbit γ-globulin (RGG) in the bile of broiler chickens fed with BSA during the first 3 d of age, or not (PBS), receiving delayed or early nutrition in experiment I. Broilers were intratracheally immunized with BSA and RGG at 21 d of age and bile was collected at 21 d post immunization. Data are presented as estimated marginal means with standard errors (SEM).

	Treatments												Fixed Effects ^1^		
	Early Nutrition PBS Fed			Early Nutrition BSA Fed			Delayed Nutrition PBS Fed			Delayed Nutrition BSA Fed					
Antigen	Mean	SEM	*n* ^2^	Mean	SEM	*n*	Mean	SEM	*n*	Mean	SEM	*n*	BSA	Feeding	BSA × Feeding
BSA	5.1	0.38	11	6.4	0.55	8	4.4	0.33	12	5.0	0.36	13	0.05	0.05	0.47
RGG	5.2	0.18	11	5.6	0.23	8	5.0	0.17	12	5.2	0.17	13	0.18	0.18	0.68

^1^ Model established *p*-values for fixed effect of BSA-feeding, feeding strategy, and their interaction. ^2^ Number of replicate broilers, housed with 4–5 broilers per pen.

**Table 2 vetsci-07-00148-t002:** Body weight and average daily gain (ADG) of broiler chickens receiving either delayed or early nutrition in experiment I. Data are presented as estimated marginal means with standard errors (SEM).

Parameter	Age (d)	Early Nutrition			Delayed Nutrition			Feeding Effect ^1^
		Mean	SEM	*n* ^2^	Mean	SEM	*n*	
Body weight (g)	0	43.8	0.39	27	44.1	0.40	26	0.60
	21	889	19.8	25	737	16.1	26	**0.001**
	24	1135	21.9	25	958	26.4	26	**0.001**
	42	3418	79.1	25	3202	78.2	26	*0.09*
Average daily gain (g/d)	0–21	40.2	0.94	25	33.0	0.76	26	**0.001**
	21–24	83	2.9	25	74	2.8	26	*0.07*
	24–42	125	3.9	25	124	3.9	26	0.79
	0–42	80	1.9	25	75	1.9	26	*0.09*

^1^ Model established *p*-values for fixed effect of feeding strategy. ^2^ Number of replicate broilers, housed with 4–5 broilers per pen.

**Table 3 vetsci-07-00148-t003:** Fold change (7 d relative to 0 d post immunization) of antibody titers (IgM, IgY) binding bovine serum albumin (BSA) or rabbit γ-globulin (RGG) in blood serum of broiler chickens fed with BSA during the first 3 d of age receiving delayed or early nutrition, kept under high (HSC) or low sanitary condition (LSC) in experiment II. Broilers were intratracheally immunized with BSA and RGG at 21 d of age, and bile was collected at 11 d post immunization. Data are presented as estimated marginal means with standard errors (SEM).

Antigen	Isotype	Early Nutrition LSC		Early Nutrition HSC		Delayed Nutrition LSC		Delayed Nutrition HSC		Fixed Effects ^1^			
		Mean	SEM	Mean	SEM	Mean	SEM	Mean	SEM	SC	Feeding	Batch ^2^	SC × Feeding
BSA	IgM	1.3	0.17	2.2	0.29	1.4	0.18	2.1	0.27	**0.01**	0.99	*0.07*	0.53
BSA	IgY	1.3	0.16	1.6	0.19	1.3	0.16	1.5	0.18	0.21	0.99	0.25	0.80
RGG	IgM	1.3	0.13	1.8	0.18	1.2	0.13	1.9	0.19	**<0.01**	0.88	**0.01**	0.76
RGG	IgY	1.3	0.13	1.7	0.17	1.2	0.12	1.4	0.14	*0.07*	0.22	**0.02**	0.49

Number of replicate broilers, housed with 5 broilers per pen, is 12 per treatment, except for EN–LSC, where *n* = 11. ^1^ Model established *p*-values for fixed effect of sanitary condition, feeding strategy, batch, and their two-way interactions. ^2^ There were no interactions between batch and sanitary conditions or feeding strategy.

**Table 4 vetsci-07-00148-t004:** IgA-binding bovine serum albumin (BSA) or rabbit γ-globulin (RGG) in bile of broiler chickens fed with BSA during the first 3 d of age receiving delayed or early nutrition kept under high (HSC) or low sanitary condition (LSC) in experiment II. Broilers were intratracheally immunized with BSA and RGG at 21 d of age and bile was collected at 11 d post immunization. Data are presented as estimated marginal means with standard errors (SEM).

Antigen	Early Nutrition LSC			Early Nutrition HSC			Delayed Nutrition LSC			Delayed Nutrition HSC			Fixed Effects ^1^				
	Mean	SEM	*n* ^2^	Mean	SEM	*n*	Mean	SEM	*n*	Mean	SEM	*n*	Feeding	SC	Batch	SC × Feeding	Batch × Feeding
BSA	7.7	0.25	32	6.0	0.24	35	7.6	0.26	29	6.0	0.24	35	0.71	**<0.001**	0.20	0.91	*0.06*
RGG	7.5	0.15	32	6.1	0.15	36	7.1	0.16	29	5.9	0.15	35	**0.03**	**<0.001**	0.37	0.72	- ^3^

^1^ Model established *p*-values for fixed effect of sanitary condition, feeding strategy, batch, and their two-way interactions. ^2^ Number of replicate broilers, housed with 4–5 broilers per pen. ^3^ Batch * Feeding effect was left out of the model as *p* > 0.10.

**Table 5 vetsci-07-00148-t005:** Bodyweight (BW), average daily gain (ADG), and feed conversion ratio (FCR) of broiler chickens receiving either delayed or early nutrition kept under high (HSC) or low sanitary condition (LSC) in experiment II. Data are presented as estimated marginal means with standard errors (SEM).

Parameter	Age	Early Nutrition LSC			Early Nutrition HSC			Delayed Nutrition LSC			Delayed Nutrition HSC			Fixed Effects ^1^				
		Mean	SE	*n* ^2^	Mean	SE	*n*	Mean	SE	*n*	Mean	SE	*n*	Feeding	SC	Batch	Feeding × SC	Batch × SC
Body weight	3	N/A ^3^			68.7	0.42	24	N/A			38.5	0.42	24	**<0.001**	-	**<0.001**	-	-
(g)	33	1633	36.2	12	2071	36.2	12	1313	36.2	12	1771	36.2	12	**<0.001**	**<0.001**	0.02	0.78	**<0.01**
Average daily	3–14	26	0.5	12	32	0.5	12	17	0.5	12	23	0.5	12	**<0.001**	**<0.001**	**<0.001**	0.73	**<0.001**
gain (g/d)	14–28	59	1.5	12	78	1.5	12	50	1.5	12	68	1.5	12	**<0.001**	**<0.001**	0.12	0.82	**0.01**
	28–33	90	2.6	12	112	2.6	12	77	2.6	12	105	2.6	12	**<0.01**	**<0.001**	0.32	0.28	**0.04**
	3–33	52	1.2	12	67	1.2	12	42	1.2	12	58	1.2	12	**<0.001**	**<0.001**	**0.03**	0.72	**0.01**
Average daily	3–14	32 ^b^	0.7	12	37 ^a^	0.7	12	21 ^d^	0.7	12	28 ^c^	0.7	11	**<0.001**	**<0.001**	**<0.001**	**0.03**	*0.07*
feed intake (g/d)	14–28	86	2.9	12	108	2.9	12	70	2.9	12	90	3.0	9	**<0.001**	**<0.001**	*0.09*	0.83	**0.04**
	28–33	149	4.8	10	173	5.0	11	121	4.8	12	160	4.8	10	**<0.001**	**<0.001**	**0.95**	0.14	0.13
	3–33	74	2.0	12	90	2.1	11	58	2.0	12	77	2.1	11	**<0.001**	**<0.001**	0.16	0.42	*0.09*
Feed conversion ratio	3–14	1.24	0.037	12	1.15	0.039	12	1.22	0.037	12	1.23	0.041	11	0.48	0.44	**<0.001**	0.21	**<0.001**
	14–28	1.44	0.017	12	1.39	0.017	12	1.39	0.017	12	1.32	0.018	9	**0.01**	**<0.01**	0.11	0.47	0.63
	28–33	1.60	0.023	10	1.54	0.022	11	1.57	0.020	12	1.51	0.022	10	0.26	**0.03**	*0.09*	0.89	0.50
	3–33	1.43	0.011	12	1.34	0.011	11	1.37	0.011	12	1.32	0.012	9	**<0.01**	**<0.001**	0.56	0.18	0.13
Corrected FCR ^4^	3–33	1.46	0.011	12	1.34	0.011	11	1.44	0.011	12	1.34	0.013	9	0.71	**<0.001**	0.36	0.23	0.01

^1^ Model established *p*-values for the fixed effect of sanitary condition, feeding, batch, and their two-way interactions. ^2^ Number of replicate pens, containing 7 (at 3 d of age) or 5 (>7 d of age) broilers per pen. ^3^ Until 3 d of age, all broilers were kept under HSC. ^4^ Feed conversion ratio standardized to 2000 g of slaughter weight.

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
