# Peer review of "Effects of Early Nutrition and Sanitary Conditions on Oral Tolerance and Antibody Responses in Broiler Chickens"

_vetsci, 2020, doi:10.3390/vetsci7040148_

Round 1
Reviewer 1 Report
Dear Authors, Please see my comments below: - Line 27 and 28. Confusing statement, Please rewrite this statement: Broilers kept under LSC versus HSC, had higher antibody levels and severely depressed growth performance. Section 2.1.starter was provided for only 7 days, why only 7 days? Normally starter is for first 14 days. Please provide diet information for each stage in a table. - Line 71-72. How the chicks transported, what condition? -Line 80. Ad libitum should be Italic. -Line 88-89 and 102-103. Two statements contradict each other. Broilers were not vaccinated in the hatchery or during the study. Broilers were vaccinated against Newcastle disease at 3 d of age. -Line 142. How the birds euthanized? Table 2. For the first 48-72h chicks can consume their yolk sac and do not require feed. I doubt it Body weight can be affected significantly. Body weight and P-values look incorrect. Also, based on number provided in the table, there were no mortality during the study even chickens were under stress because of injection or collecting blood. Table 3. Most of numbers are same, you need to provide 3 decimals for all data. Table 4. 2 decimals. Table 5. Again BW data. Chicken looked affect with other factors because at age 3 their BW is already significantly lower. Table 5 has superscripts but not other tables or data. Regards,Author Response
Dear reviewer,
Attached you will find our responses to the comments.

Reviewer 2 Report
The study is well designed and the results well described. I have no suggestions for improving it. I very much praise the importance of publishing no-effect data, when the study is serious and accurate.
Author Response
Dear reviewer,
Attached you will find our responses to the comments.

Reviewer 3 Report
line 37 - can you quantify the percentage of hatchlings that experience this delay in accessing food ? to give the reader a feel for how common this is
line 70 - my understanding was that Male broilers were not kept or used for meat production - are these animals surplus ? do you think that there might be sex differences in the responses you're reporting ? if so could this be clarified and statements on applicability of these results if males aren't normally part of the production process
line 76 - why was 36 degrees chosen ? this seems like a high temperature normally range for broilers is around 25 degrees
methods are clear and well described
Author Response

(The authors gave the same response as above.)

Round 2
Reviewer 1 Report
Dear Authors,
Thank you for the revised version. I have no comment on the improved version.
Regards,